# Cytokinin Modulates Cellular Trafficking and the Cytoskeleton, Enhancing Defense Responses

**DOI:** 10.3390/cells10071634

**Published:** 2021-06-29

**Authors:** Lorena Pizarro, Daniela Munoz, Iftah Marash, Rupali Gupta, Gautam Anand, Meirav Leibman-Markus, Maya Bar

**Affiliations:** 1Institute of Agri-Food, Animal and Environmental Sciences, Universidad de O’Higgins, Rancagua 2820000, Chile; daniela.munoz@uoh.cl; 2Department of Plant Pathology and Weed Research, Institute of Plant Protection, ARO, Volcani Institute, Rishon LeZion 7505101, Israel; iftah.marash@mail.huji.ac.il (I.M.); rupaligupta862@gmail.com (R.G.); gautaming@gmail.com (G.A.); meiravleibman@gmail.com (M.L.-M.); 3School of Plant Sciences and Food Security, Faculty of Life Sciences, Tel Aviv University, Tel Aviv 6997801, Israel

**Keywords:** cytokinin, endocytosis, cytoskeleton, actin, plant immunity, induced resistance

## Abstract

The plant hormone cytokinin (CK) plays central roles in plant development and throughout plant life. The perception of CKs initiating their signaling cascade is mediated by histidine kinase receptors (AHKs). Traditionally thought to be perceived mostly at the endoplasmic reticulum (ER) due to receptor localization, CK was recently reported to be perceived at the plasma membrane (PM), with CK and its AHK receptors being trafficked between the PM and the ER. Some of the downstream mechanisms CK employs to regulate developmental processes are unknown. A seminal report in this field demonstrated that CK regulates auxin-mediated lateral root organogenesis by regulating the endocytic recycling of the auxin carrier PIN1, but since then, few works have addressed this issue. Modulation of the cellular cytoskeleton and trafficking could potentially be a mechanism executing responses downstream of CK signaling. We recently reported that CK affects the trafficking of the pattern recognition receptor LeEIX2, influencing the resultant defense output. We have also recently found that CK affects cellular trafficking and the actin cytoskeleton in fungi. In this work, we take an in-depth look at the effects of CK on cellular trafficking and on the actin cytoskeleton in plant cells. We find that CK influences the actin cytoskeleton and endomembrane compartments, both in the context of defense signaling—where CK acts to amplify the signal—as well as in steady state. We show that CK affects the distribution of FLS2, increasing its presence in the plasma membrane. Furthermore, CK enhances the cellular response to flg22, and flg22 sensing activates the CK response. Our results are in agreement with what we previously reported for fungi, suggesting a fundamental role for CK in regulating cellular integrity and trafficking as a mechanism for controlling and executing CK-mediated processes.

## 1. Introduction

The plant hormone cytokinin (CK) is central to plant life, regulating many processes including embryogenesis, cell division and differentiation, stem cell maintenance, growth and branching of roots and shoots, leaf senescence, nutrient balance, and stress tolerance. CHASE domain-containing AHK (Arabidopsis histidine kinase) receptors, which are located primarily in the endoplasmic reticulum (ER), perceive CK and commence the resultant signaling cascades, which mediate CK function [1,2,3]. Based on the subcellular localization of AHKs, the endoplasmic reticulum (ER) has been considered to be the primary CK perception site [4,5]. Upon perception through AHKs, AHPs (Arabidopsis histidine phosphotransferases) transfer a phosphate from AHKs to downstream type B ARRs (Arabidopsis response regulators), initiating transcriptional reprogramming as a result of CK perception [6]. CK signaling can also start at the plasma membrane (PM) [7]. Extracellular CKs were shown to bind to cell-surface receptors and elicit signaling [8]. ER-localized CK receptors were also demonstrated to be transported to the PM [9]. In that work, AHK4 was shown to reside in the PM and in vesicles, in addition to the ER. Moreover, CK was shown to accumulate in BFA compartments, attesting to AHKs being trafficked between the ER and PM on endocytic vesicles. Thus, initiation sites of cytokinin perception occur at both the PM and the ER, and CK receptors undergo endosomal trafficking between these locales. It has been suggested that the dual CK response locales may contribute to the flexibility of the CK response [8].

CK regulates some developmental processes through crosstalk with the auxin pathway. CK was previously shown to regulate the transcription of auxin pathway genes [10,11,12]. Importantly, CK was also demonstrated to modulate auxin activity by regulating the endocytic trafficking of the auxin carrier PIN1 [13]. CK regulates the recycling of the auxin efflux carrier PIN1 to the plasma membrane, by directing it to the vacuole for degradation, in a rapid regulatory pathway that does not require transcriptional reprogramming [13]. Thus, through the endocytic redirection of PIN1, CK controls auxin fluxes that are required for lateral root organogenesis, and it was postulated that this endocytic regulation may also be involved in other CK-regulated developmental processes. However, CK was found not to influence the stability of two additional membranal auxin carrier proteins, prompting the authors to suggest that CK affects trafficking in a protein-specific manner [13].

CKs were shown to mediate disease resistance through induction of host immunity. Transgenic plants with high CK levels had increased resistance to *Pseudomonas syringae* [14], while transgenes with low levels of CK or Arabidopsis histidine kinase (AHK) receptor mutants displayed enhanced pathogen susceptibility [14,15]. CKs were also found to mediate enhanced resistance in tobacco [16]. In Arabidopsis, it was suggested that CK-mediated resistance depends on salicylic acid (SA) [14], and that CK signaling enhances SA-mediated immunity [17]. In tobacco, an SA-independent, phytoalexin-dependent mechanism was reported [16]. We recently reported that CK induces systemic immunity in tomato, promoting resistance to fungal and bacterial pathogens in an SA- and ethylene (ET)-dependent mechanism [18,19]. Until recently, other than the evidence that SA is required for CK-induced immunity [14,15,17], and that phytoalexins are involved [16], no additional cellular mechanisms related to CK-mediated immunity had been reported.

Pattern recognition receptors (PRRs) are the first line of defense and immune activation in plant cells. Since CK was shown to modulate endocytic trafficking of PIN1 in its regulation of auxin [13], in our recent work, we examined whether CK might also affect the trafficking of immune receptors as a possible mechanism for promoting immune responses and disease resistance. We were able to show that CK modulates the cellular trafficking of the PRR LeEIX2, which mediates immune responses to Xyn11 family xylanases [18]. We found that CK enhances both the endosomal presence and the vesicular size of LeEIX2 endosomes, without affecting the total cellular content of the proteins [18]. Furthermore, asking whether this CK-mediated enhancement of PRR endosomal presence acts as a mechanism for increased disease resistance, we used a SlPRA1A-overexpressing line, which has a decreased presence of receptor-like protein (RLP)-type PRRs in the cell plasma membrane, due to receptor degradation [20]. Indeed, we found that in plants overexpressing SlPRA1A, which have decreased levels of RLP-type PRRs, CK-mediated disease resistance to *Botrytis cinerea* is compromised [18].

Recently, we have found that CK directly affects the growth, development, and virulence of fungal plant pathogens [21]. *Botrytis cinerea* (*Bc*) growth, sporulation, and spore germination were all inhibited by CK, with some fungal developmental processes being inhibited by plant physiological CK concentrations [21]. We found similar effects in a variety of plant pathogenic fungi. We also found that CK affects both budding and fission yeast in a similar manner. Transcriptome profiling revealed that cell cycle and DNA replication, cytoskeleton integrity, and endocytosis are all inhibited in *Bc* by CK [21]. We directly confirmed these results, demonstrating that CK affects the cell cycle and DNA replication, cytoskeleton distribution, and cellular trafficking in fungi and yeasts [21]. CK interfered with actin localization in growing hyphae and lowered endocytic rates and endomembrane compartment sizes, likely underlying the reduced growth and development we observed [21].

In this work, we take an in-depth look at the effects of CK on cellular trafficking and on the cytoskeleton in plant cells, both in the context of plant immunity and in general. We have found that in addition to the xylanase receptor-like protein (RLP) LeEIX2, CK also affects the distribution of the flagellin receptor-like kinase (RLK) flagellin-sensing 2 (FLS2), increasing its presence in the plasma membrane. FLS2, first characterized in Arabidopsis, acts as the PRR for the bacterial PAMP (pathogen-associated molecular pattern) flagellin [22] in several plant species [23,24,25]. Furthermore, CK enhances the cellular response to the 22-amino-acid, flagellin-derived peptide flg22, and flg22 sensing activates the CK response. Examining cellular trafficking compartments and the cytoskeleton in general, we also show that CK affects endosome distribution and increases the amount of endomembrane compartments. CK also caused disorganization and reduction in actin filaments, but not in tubulin. Our results are in agreement with what we previously reported for fungi, suggesting a fundamental role for CK in regulating cellular integrity and trafficking as a mechanism for controlling and executing CK-mediated processes.

## 2. Results

### 2.1. CK Affects PRR Trafficking

We previously reported that CK modulates the trafficking of the PRR LeEIX2 and enhances its defense signaling, and that PRR-RLPs are required to achieve CK-mediated disease resistance [18]. To examine whether this is a general effect on PRRs, we tested whether CK influences the signaling of the LRR-receptor-like kinase (RLK)-type PRR FLS2. FLS2 is very different from LeEIX2 in its structure, specificity, and regulation [20,22,23]. Figure 1 demonstrates that CK affects cellular distribution of FLS2, increasing its presence at the PM.

### 2.2. CK Response Is Activated during Immunity Elicitation

We and others have previously demonstrated that pathogenesis processes can activate the CK pathway [17,18,19], raising the possibility that CK pathway activation might be required for pathogen resistance. Interestingly, we previously found that *B. cinerea* infection activates the synthetic CK response promoter TCSv2 [18]. To test whether defense elicitation activates the CK pathway, we examined whether TCSv2 could respond to flg22 elicitation. As seen in Figure 2, flg22 treatment in Arabidopsis roots significantly enhanced TCSv2-driven Venus expression, indicating that flg22 activates the CK response.

Our previous work has shown that in addition to affecting LeEIX2 trafficking, CK enhances defense signals elicited by its ligand, EIX [18]. Since we observed that CK also affected FLS2 cellular distribution, increasing its presence at the PM, we tested whether CK also enhanced defense signaling elicited by flg22. As seen in Figure 3, CK strongly enhances flg22 elicited reactive oxygen species (ROS) production. CK also has a small but significant inductive effect on ROS production on its own (Figure 3B). The CK trans-zeatin was previously reported to induce ROS production in Arabidopsis guard cells [26].

### 2.3. CK Treatment Modulates Cellular Trafficking

As a developmental hormone, CK regulates various processes that require growth and rapid membrane modeling. These processes could be controlled through cellular trafficking. We have recently found that in fungi, CK attenuates cellular trafficking and causes mislocalization in the actin cytoskeleton [21]. Interestingly, investigation of the effect of CK on the cellular trafficking of the auxin carrier PIN1 showed that CK does not affect the cellular localization of two additional auxin carriers [13].

To test whether the effect of CK on cellular trafficking is general or specific to defense receptors, we examined the distribution of ARA6 and FM-4-64 endosomes following CK treatment. As seen in Figure 4, CK treatment significantly increases the density and size of both FM4-64 compartments (Figure 4A–C) and ARA6 endosomes (Figure 4A,D,E). ARA6 is considered to be a late endosome marker [27], while FM4-64 is present in all endomembrane compartments at the timepoint tested [28].

### 2.4. CK Treatment Affects the Cellular Cytoskeleton

Growth requires changes to both the composition and the orientation of the cytoskeleton [29]. Furthermore, observed effects on vesicular trafficking can relate to the cytoskeleton, since trafficking relies on an intact cytoskeleton [30]. We previously demonstrated that CK can affect F-actin distribution in fungi [21]. We therefore examined whether CK treatment could affect the plant cytoskeleton. Figure 5 shows that CK affects actin distribution, reducing the quantity and organization of actin in the cell (Figure 5A–C). CK did not affect tubulin (Figure 5D–F).

## 3. Discussion

CK mediates plant immunity and disease resistance [14,15,16,17,18,19]. Traditionally a plant growth hormone with various developmental roles, CK supports juvenility and delays senescence [31,32], indicating that perhaps commonalities beyond the execution of the cell death program may exist between age-related cell death and pathogen-related cell death, both of which CK can delay or prevent.

In addition to its role in controlling auxin distribution by regulating its trafficking [13], we recently reported that CK can influence cellular trafficking in plants—where it enhances defense signaling of EIX through modulation of the trafficking of its PRR LeEIX2 [18]—and in fungi, where CK treatment lowers growth and development by attenuating trafficking [21]. Furthermore, we observed that CK-mediated disease resistance was lost in a transgenic line with reduced PRR expression [18]. These reports suggest that control of cellular trafficking could be a general mechanism by which CK exerts its varied effects on development and defense.

In this work, we affirmed the role we previously reported for CK in controlling receptor trafficking, demonstrating that CK enhances flagellin signaling (Figure 2), possibly by increasing the presence of its receptor—FLS2—in the membrane (Figure 1), from which it may transmit part of its defense signals [33]. Interestingly, though ligand-activated FLS2 is endocytosed after about 80–90 min [34], ROS responses occur within 5 min (Figure 3), suggesting that an increase in FLS2 presence at the membrane could underlie the increase in ROS observed upon CK treatment. Furthermore, we observed that flagellin activates CK response (Figure 2). Flagellin was previously reported to increase expression of the CK receptor AHK3 and the CK response regulator ARR2 in Arabidopsis guard cells [26]. We also previously observed a similar mode of action—activation of the CK pathway—in the pathogenesis of *B. cinerea* [18] and bacterial pathogens [19].

The fact that CK enhances signals emanating from both the fungal response RLP LeEIX2 [18] and the bacterial response RLK FLS2 (Figure 1 and Figure 2), and augments defense elicited by both of their respective ligands, EIX [18]—which is derived from *Trichoderma*, which mostly signals through the jasmonic acid pathway [35], and flg22 (Figure 2), which mostly signals through the salicylic acid pathway [36]—indicates that the modulation of the cellular distribution of PRRs is likely a general mechanism by which CK affects cellular defense.

Our recent report [21] observed attenuation of cellular trafficking and disorganization of the actin cytoskeleton in response to CK treatment in *B. cinerea*, suggesting that perhaps control of cellular trafficking and cytoskeletal integrity could be a general mechanism employed to execute CK-mediated responses in plants as well. We examined endocytosis and actin distribution in response to CK treatment in Arabidopsis and *N. benthamiana*. We found that CK significantly increased the size and density of endomembrane compartments, as observed with both the general FM-4-64 dye and the late-endosome-specific ARA6 (Figure 4). Interestingly, the effects of CK on endomembrane compartments did not appear to be dose-dependent in most examined cases, with a cutoff of 1 μM observed, though this effect could be specific to 6-BAP in Arabidopsis, and testing of additional CK concentrations is required in order to definitively make this assertion.

Taken together, our results indicate that one of the early mechanisms for the activation of CK-mediated plant immunity is likely through cellular trafficking. CK may enhance defense signals, both by specifically increasing PRR occupancy in the cellular compartments required for signaling, and by increasing cellular trafficking compartments in general. flg22 is known to induce the entry of FLS2 to endosomes. flg22-induced FLS2-GFP endosomes were shown to co-localize to ARA6 compartments, though there was no difference in the amount of ARA6 endosomes at time points between 30 min and 105 min after flg22 application [37]. This could support the notion that the increased immunity observed upon flg22 and CK co-treatment (Figure 3) stems both from the increased presence of FLS2 in signaling compartments due to flg22 activation, and from a general increase in the available amount of endomembrane compartments, due to CK-mediated activity.

We found that CK inhibits the actin cytoskeleton, though not the tubulin cytoskeleton, in plant cells (Figure 5). We observed similar inhibition of the actin cytoskeleton in *B. cinerea*, underlying the mechanism for growth and development inhibition in fungi [21]. Actin inhibition likely also underlies CK-mediated growth inhibition observed in plants, as exemplified in the inhibition of root growth [38], and also perhaps in processes that maintain plant juvenility, e.g., inhibition of changes to cellular trafficking or degradation of the cytoskeleton required for senescence and cell death. The cytoskeleton is known to be an important component of cellular trafficking. The effect of CK on the actin cytoskeleton could result in altered trafficking rates and paths for endomembrane compartments—which could, in turn, affect the distribution of defense-related cargo and result in altered defense signaling.

Interestingly, our findings on the effects of CK on cellular trafficking differ somewhat between plants (Figure 4) and fungi [21]. While in both cases we observed a decrease in both the content and organization of actin filaments, in fungi—in contrast to plants—we observed a decrease in endomembrane compartments upon CK treatment. These differences could relate to differences in both the assay methodologies and the cell biology attributes of different species. Notably, the endocytosis pathway was significantly downregulated upon CK treatment in *B. cinerea* [21].

In summary, we found that CK regulates actin distribution, endocytosis, and PRR trafficking. A model summarizing this work is provided in Figure 6. This is an interesting starting point for future work, which will elucidate how the dual role of CK in growth and defense is regulated, and which additional proteins and feedback circuits are involved, perhaps also elucidating how some of these functionalities evolved in both plants and fungi.

## 4. Materials and Methods

### 4.1. Plant Growth Conditions

*Nicotiana benthamiana* and *Solanum lycopersicum* cv. M82 were grown from seeds in soil (Green Mix; Even Ari, Ashdod, Israel) in a growth chamber at 24 °C, under long-day conditions (16 h:8 h, light:dark).

*A. thaliana* cv. Columbia transgenic plants overexpressing *pTCS::3XVENUS* [39] or *pARA6::ARA6-Venus* [27] were germinated on 1/2 MS (Duchefa M0225) agar plates containing 50 ug/mL kanamycin, and grown upright in a growth chamber at 22 °C under short-day conditions (8 h:16 h, light:dark).

### 4.2. Transient Expression

Four-week-old *N. benthamiana* plant leaves were abaxially infiltrated with a needleless syringe, with an *Agrobacterium tumefaciens* (strain GV3101) harboring the following previously described constructs: *pAtFLS2::AtFLS2-3xmyc-GFP* [40] (Addgene plasmid#86157; http://n2t.net/addgene:86157, accessed on 31 May 2021; RRID:Addgene_86157), *pCMU-ACTLr* (AtUBQ10::F-actin LifeAct-mCherry; Addgene plasmid #61193; http://n2t.net/addgene:61193, accessed on 31 May 2021; RRID:Addgene_61193) [41], and *pCMU-MTUBr* (AtUBQ10::mCherry-MAP4-MBD; Addgene plasmid #61196; http://n2t.net/addgene, accessed on 31 May 2021; RRID:Addgene_61196) [41]. Treatments were applied 40 h after infiltration. Transgenic lines and constructs used in this study are detailed in Table 1.

### 4.3. Chemical Treatments

The CK 6-benzylaminopurine (6-BAP, Benzyladenine, sigma) was dissolved in 10 mM NaOH. *pARA6:ARA6-Venus* Arabidopsis plants were grown in 0.5X MS-agar media and transferred 3 days after germination to 0.5X MS-agar media supplemented with 6-BAP at indicated concentrations. *pTCS::3XVENUS* Arabidopsis plants, 5 days after germination, were incubated for 16 h in 0.5X MS media supplemented with 6-BAP at indicated concentrations (Figure 2 and Figure 4). In the transient expression assays, 6-BAP water solutions were infiltrated into *N. benthamiana* leaves, and images were captured 6 h after infiltration (Figure 1 and Figure 5). flg22 (1 μM, PhytoTech Labs #P6622) was dissolved in DMSO and applied as described for CK.

FM-4-64 (Invitrogen # T13320) was dissolved in water and applied to Arabidopsis roots in solution at 5 μM. Arabidopsis roots were incubated with this solution for 5 min on ice prior to live-cell imaging.

### 4.4. Confocal Microscopy

Confocal microscope images were acquired using a Zeiss LSM780 confocal microscope with Objective C-Apochromat 40 ×/1.2 W Corr M27 (Figure 1, Figure 2, and Figure 5) or Objective C-Apochromat 63×/1.2 W Corr (Figure 4). GFP was excited using a laser of 488 nm (5% power), and emission was collected in the range of 493–535 nm. mCherry fluorescence was excited using a laser of 561 nm (3% power), and emission was collected in the range of 588–641 nm. FM4-64 was excited using a laser of 514 nm (3% power), and emission was collected in the range of 650–750 nm. Images of 8 bits were acquired using a pixel dwell time of 1.27 μs, pixel averaging of 4, and pinhole of 1 airy unit. Image analysis was performed using Fiji-ImageJ with the raw images [42]. Endosome density and size measurements were conducted with the 3D Object counter tool, pixel intensity was measured using the measurement analysis tool, cytoskeleton organization was measured using the skew tool, and occupancy was assessed using the 3D Object tool [42].

### 4.5. Reactive Oxygen Species Burst Measurement

ROS measurement was performed as previously described [43]. Leaf disks of 0.5 cm in diameter were harvested from leaves 4 to 6 of 5–6-week-old M82 plants, treated for 12 h with mock (1 mM NaOH) or CK (100 μM 6-benzylaminopurine). Disks were floated in a white 96-well plate (SPL Life Sciences, Korea) containing 250 μL distilled water for 4–6 h at room temperature. After incubation, water was removed, and an ROS measurement reaction containing 1 μM flg22 was added. Light emission was immediately measured using a luminometer (Spark, Teacan, SL).

### 4.6. Data Analysis

Differences between two groups were analyzed for statistical significance using an unpaired two-tailed *t*-test, with Welch’s correction where samples were found to have unequal variances, or with Holm–Sidak correction where multiple *t*-tests were applied. Differences among three groups or more were analyzed for statistical significance with a one-way ANOVA, or with a Kruskal–Wallis test. For ANOVA analyses, regular ANOVA was used for groups with equal variances, and Welch’s ANOVA for groups with unequal variances. When a significant result for a group was returned, differences between the means of different samples in the group were assessed using a post-hoc test. Tukey’s test was employed for ANOVA conducted on samples with equal variances, Dunnett’s test was employed for ANOVA conducted on samples with unequal variances, and Dunn’s multiple comparisons post-hoc test was employed for Kruskal–Wallis tests. All statistical analyses were conducted using Prism9.

## Figures and Tables

**Figure 1 cells-10-01634-f001:**
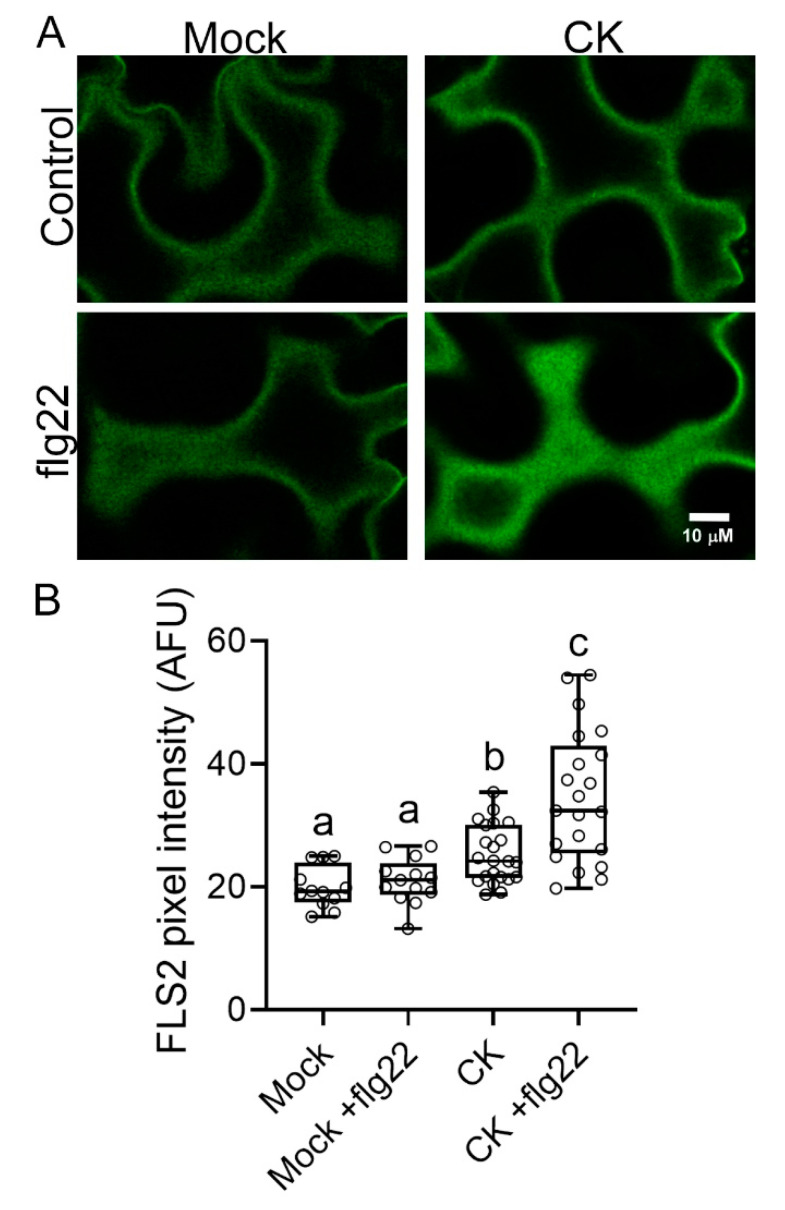
Cytokinin increases the membranal presence of the PRR FLS2. *N. benthamiana* epidermal cells transiently expressing FLS2-GFP were mock treated, or treated with cytokinin (CK, 100 µM 6-benzylaminopurine), for 4 h, and subsequently treated with flg22 (30 min). Membranal plane images (1 µm) were captured in 3 experiments, and images were analyzed using Fiji-ImageJ. (**A**) Representative confocal microscopy images. Contrast was uniformly adjusted; scale bar = 10 µm. (**B**) Twelve images were analyzed. Data are presented as boxplots with inner quartile ranges (boxes), outer quartile ranges (whiskers), and medians (lines in boxes), all points displayed. Letters represent statistical significance in one-way ANOVA with Dunnett’s post-hoc test (*p* < 0.035).

**Figure 2 cells-10-01634-f002:**
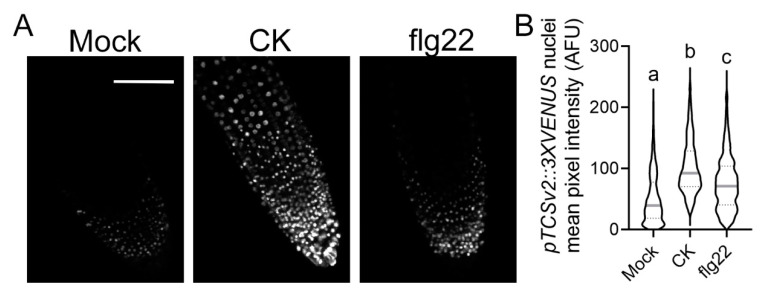
flg22 activates cytokinin response. Transgenic *A. thaliana* roots stably expressing the cytokinin (CK) response marker *pTCSv2::3XVENUS* were treated with mock (1 mM NaOH), CK (100 µM 6-benzylaminopurine), or flg22 (1 µM), for 12 h. Images were captured in 3 experiments using at least 15 plants per treatment, and analyzed using Fiji-ImageJ; scale bar = 100 µm. (**A**) Representative confocal microscopy images. Contrast was uniformly adjusted; scale bar = 100 µm. (**B**) Images from at least 15 plants were analyzed. Mean fluorescence intensity for each nucleus in each image was quantified, N > 770 nuclei. Data are presented as violin plots—solid gray lines indicate medians, and dotted lines indicate quartile ranges. Letters represent statistical significance in a Kruskal–Wallis test with Dunn’s multiple comparisons post-hoc test (*p* < 0.0001).

**Figure 3 cells-10-01634-f003:**
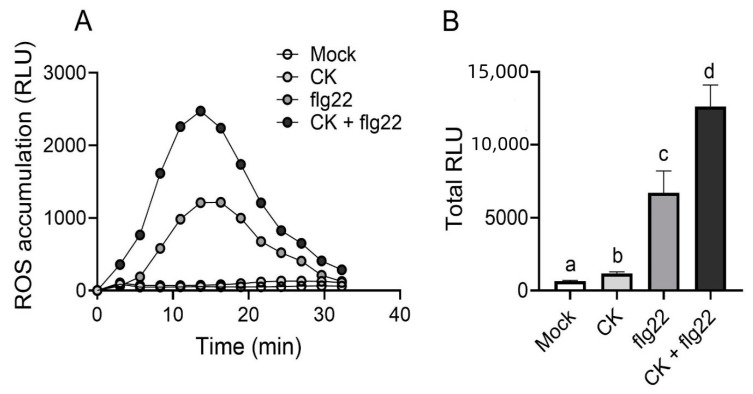
Cytokinin enhances flg22-mediated defense. *S. lycopersicum* cv. M82 leaves were treated for 12 h with mock or with cytokinin (CK, 100 µM 6-benzylaminopurine). After 12 h, leaf discs were harvested and treated with flg22 (1 µM). Reactive oxygen species (ROS) production was measured every 2.5 min for 35 min, using the HRP-luminol method. (**A**) Kinetics of ROS accumulation are plotted; RLU = relative light units; N = 24. (**B**) Total ROS production, area under the graph in A. Average ± SEM is shown, N = 24. Different letters represent statistically significant differences in one-way ANOVA, *p* < 0.0001, with a Dunnett’s post-hoc test, *p* < 0.044.

**Figure 4 cells-10-01634-f004:**
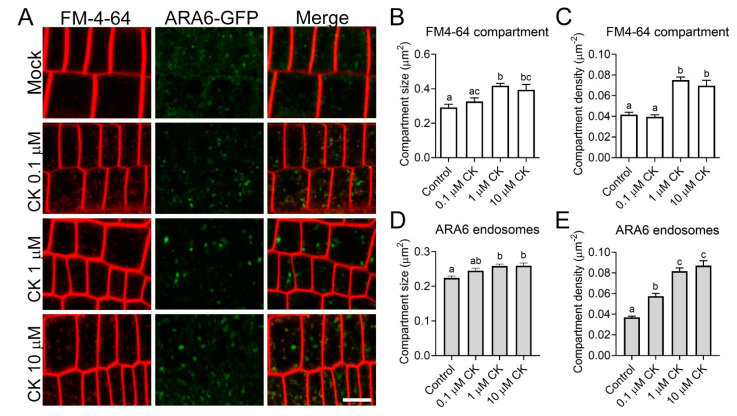
Cytokinin increases endosomal size and density. Transgenic *A. thaliana* root roots stably expressing the endosomal marker ARA6-GFP were grown vertically on 0.5X MS-agar plates, mock or supplemented with cytokinin (CK, 6-benzylaminopurine) at indicated concentrations, and stained with 5 µM FM-4-64 for 5 min on ice. Images were captured in 4 experiments using at least 10 plants per treatment, and analyzed using Fiji-ImageJ. (**A**) Representative confocal microscopy images. Contrast was uniformly adjusted; scale bar = 10 µm. (**B**–**D**) A total of at least 40 cells from at least 10 images per treatment were captured in 4 separate experiments and analyzed. Compartment size and density for FM4-64 (**B**,**C**) and ARA6-GFP (**D**,**E**) are presented as mean ± SEM. Letters represent statistically significant differences in a one-way ANOVA with Tukey’s post-hoc test. (**B**) N > 425, *p* < 0.0076. (**C**) N > 40, *p* < 0.0001. (**D**) N > 500, *p* < 0.0037. (**E**) N > 60, *p* < 0.0001.

**Figure 5 cells-10-01634-f005:**
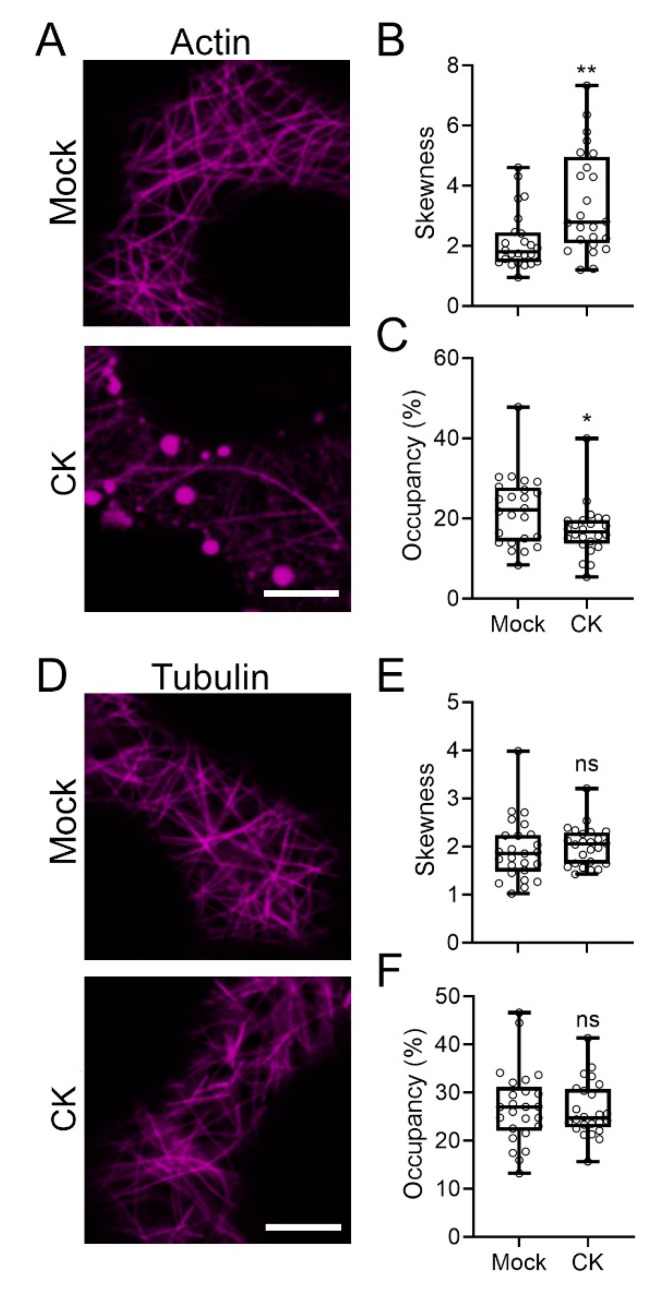
Cytokinin decreases actin organization and density. *Nicotiana benthamiana* epidermal cells transiently expressing Actin-mCherry (**A**–**C**) or mCherry-Tubulin (**D**–**F**) were mock treated, or treated with cytokinin (CK, 100 µM 6-benzylaminopurine) for 4 h. Membranal plane images (1 µm) were captured in 3 experiments, and images were analyzed using Fiji-ImageJ. (**A**,**D**) Representative confocal microscopy images. Contrast was uniformly adjusted; scale bar = 10 µm. (**B**,**C**,**E**,**F**) 25 images per treatment were analyzed. Box plots represent inner quartile ranges (boxes), outer quartile ranges (whiskers), and medians (lines in boxes), all points shown. Asterisks represent statistically significant differences in an unpaired two-tailed *t*-test with Welch’s correction, N = 25. (**B**) ** *p* = 0.0031. (**C**) * *p* < 0.028. (**E**,**F**) ns = not significant.

**Figure 6 cells-10-01634-f006:**
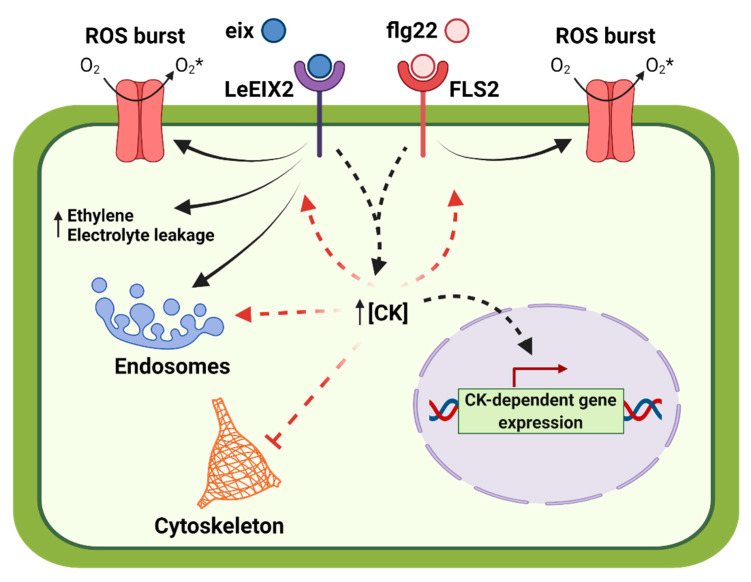
Model for the effects of cytokinin on plant cell compartments and defense signaling. Cytokinin (CK) increases endosomal compartment size and density, and decreases actin content and organization. CK promotes defense signaling mediated by both FLS2 and LeEIX2, resulting in increases in reactive oxygen species (ROS) in response to flg22 and EIX, as well as increases in ethylene and electrolyte leakage in response to EIX. CK also primes ROS on its own, and induces gene expression. LeEIX2 and FLS2 elicitor activation induces CK signaling pathways, increasing CK levels and/or activating CK-dependent gene expression. Model created with BioRender.com, accessed on 31 May 2021.

**Table 1 cells-10-01634-t001:** Transgenic lines and constructs used in this study.

Line/Construct	Source	Organism/Use	Fig#
*pAtFLS2::AtFLS2−3xmyc-GFP* [40]	Addgene plasmid#86157; http://n2t.net/addgene:86157, accessed on 31 May 2021; RRID:Addgene_86157	Transient expression in *N. benthamiana*	1
*pTCS::3XVENUS* [39]	David Weiss, Faculty of Agriculture, HUJI.	Stable transgenic *A. thaliana*, col. background	2
*pARA6:ARA6-Venus* [27]	Takashi Ueda.	Stable transgenic *A. thaliana*, col. background	4
*pCMU-ACTLr* [41]	Addgene plasmid #61193; http://n2t.net/addgene:61193, accessed on 31 May 2021;	Transient expression in *N. benthamiana*	5
*pCMU-MTUBr* [41]	Addgene plasmid #61196; http://n2t.net/addgene:, accessed on 31 May 2021; RRID:Addgene_61196	Transient expression in *N. benthamiana*	5

## Data Availability

The authors declare that the data supporting the findings of this study are available within the paper. Raw data are available from the corresponding author upon reasonable request.

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
