# Peer review of "Cytokinin Modulates Cellular Trafficking and the Cytoskeleton, Enhancing Defense Responses"

_cells, 2021, doi:10.3390/cells10071634_

Round 1

Reviewer 1 Report

Comments to manuscript Cells-1263327

Interesting topic: research on the effects of CKs on plant immune responses is a new study area. Manuscript is relatively well written, however there are several issues to be addressed before publication.

  1. Results section contains two apparently unrelated parts, (i) the effects of CK on the pathogenesis response machinery, and (2) the effects of CK on endosome formation and cytoskeletal distribution. These two topics are very important and at least the second one needs more detailed investigation. Although intracellular trafficking of PR-related molecules is suggested to be regulated by cytokinin via cytoskeleton mediated vesicular traffic, the direct relationship between these two events needs further investigation.

  1. 3: ROS accumulation should have been followed in mock- and CK treatments without flagellin as well: ROS production can be time-dependent even in controls, therefore these should be shown as well. Fig. 3b: what was the length of treatments for these ROS levels?

  1. The Results section contains many discussion-like interpretations. A combined Results and Discussion section would be beneficial.

  1. One of my main concerns is that different investigations were performed with different plant species. The Authors should make experiments on all model plants involved in the study at least where possible, e.g. assay of ROS content in the presence of CK/flagellin in wt plants to make sure that similar responses can be observed in different species. On the other hand we understand that for using transgenic lines for live cell imaging, different fusion protein constructs were probably available only in different species, but if this is the case, this should be clarified.

Abbreviations: please spell out at first mention the terms like FLS2

Minor typing/spelling corrections are indicated in the annotated manuscript file attached.

Author Response

  1. Results section contains two apparently unrelated parts, (i) the effects of CK on the pathogenesis response machinery, and (2) the effects of CK on endosome formation and cytoskeletal distribution. These two topics are very important and at least the second one needs more detailed investigation. Although intracellular trafficking of PR-related molecules is suggested to be regulated by cytokinin via cytoskeleton mediated vesicular traffic, the direct relationship between these two events needs further investigation.

Thanks for this comment. We believe there is a direct connection between the general effect of CK on the cell, and its effect on defense. We have tried to make this clearer in the revised manuscript.

  1. 3: ROS accumulation should have been followed in mock- and CK treatments without flagellin as well: ROS production can be time-dependent even in controls, therefore these should be shown as well. Fig. 3b: what was the length of treatments for these ROS levels?

Controls were added to the assay and are now included in Figure 3. CK does have a small but significant effect on ROS production on its own, as was previously reported in Arabidopsis. We added this information to the text. The graph in panel 3B is the quantification of all the "ROS" generated in 3A in that time frame, i.e., the total area underneath the graph. We added a comment to this effect to the figure legend.

  1. The Results section contains many discussion-like interpretations. A combined Results and Discussion section would be beneficial.

We will consult with the editor on this point, thanks.

  1. One of my main concerns is that different investigations were performed with different plant species. The Authors should make experiments on all model plants involved in the study at least where possible, e.g. assay of ROS content in the presence of CK/flagellin in wt plants to make sure that similar responses can be observed in different species. On the other hand we understand that for using transgenic lines for live cell imaging, different fusion protein constructs were probably available only in different species, but if this is the case, this should be clarified.

ROS production in response to CK was previously demonstrated in Arabidopsis, as detailed above. This information was added to the text. Indeed, different constructs and transgenic lines were available for the study- a Table clarifying this and adding information was included in the methods sections of the revised manuscript.

Abbreviations: please spell out at first mention the terms like FLS2

Corrected, thanks.

Minor typing/spelling corrections are indicated in the annotated manuscript file attached.

Apologies, there was a file attached but we could not find any marked up corrections in it.

Reviewer 2 Report

Dear Editor/Authors
The article presented for review contains interesting data confirming the importance of cytokines in developmental processes, and provides new data about the CK- stimulated regulation of cellular integrity and mobility. The authors chose one of the cytokinins, i.e. 6-benzylaminopurine, often used as a representative of CKs in studies of external stimulation of cellular ractions. 
The experiments are carried out correctly, the results are reliably documented in the form of photos and figures, which include statistical analysis. The authors placed annotations under the figures suggesting the next "stage" of the analysis, what I consider an interesting element of the work, explaining to the reader the sequence of experiments. Based on the obtained data, a discussion is presented to summarize the studies.
The obtained results and their discussion confirm the proposed by  Authors the possibility of CK mediation in the early stages of the immune mechanism through cellular movement. 
I propose to accept the received manuscript for publicationh.However, I suggest that the title of the work should be more appropriate to the presented research. 

Author Response

The article presented for review contains interesting data confirming the importance of cytokines in developmental processes, and provides new data about the CK- stimulated regulation of cellular integrity and mobility. The authors chose one of the cytokinins, i.e. 6-benzylaminopurine, often used as a representative of CKs in studies of external stimulation of cellular ractions.

The experiments are carried out correctly, the results are reliably documented in the form of photos and figures, which include statistical analysis. The authors placed annotations under the figures suggesting the next "stage" of the analysis, what I consider an interesting element of the work, explaining to the reader the sequence of experiments. Based on the obtained data, a discussion is presented to summarize the studies.

The obtained results and their discussion confirm the proposed by  Authors the possibility of CK mediation in the early stages of the immune mechanism through cellular movement.

I propose to accept the received manuscript for publicationh.However, I suggest that the title of the work should be more appropriate to the presented research.

Many thanks to the reviewers for their efforts. The title was amended, thanks.

Reviewer 3 Report

The manuscript entitled “The effect of cytokinin on cellular trafficking in plant defense and beyond” by L. Pizarro L, D. Munoz, I. Marash, R. Gupta, G. Anand, M. Leibman-Markus and M. Bar investigates how cytokinin affects cellular trafficking, actin cytoskeleton in planta and how it could affect plant defense using flg22 to study to the responses of the key PRR FLS2.

The method involved the use of transgenic plants with key markers sensitive to cytokinin:

  1. the use of N. benthamiana epidermal cells transiently expressing FLS2-GFP to measure its response to cytokinin and flg22;
  2. A. thaliana plants where the roots stably expressed the cytokinin (CK) response marker pTCSv2::3XVENUS and using pixel intensity to measure its response to cytokinin and flg22;
  3. ROS measurements in S. lycopersicum cv. M82 leaves in response to cytokinin and flg22;
  4. A. thaliana plants where the roots stably expressed the endosomal marker ARA6-GFP and measuring the responses to different concentration of cytokinin only and FM4-64 (which binds to plasma membranes (including vesicles);
  5. The use of N. benthamiana expressing Actin-mCherry or mCherry-Tubulin and measuring the responses to cytokinin only.

The authors concluded by the key observations in experiments using these constructs and techniques/methods that cytokinin affected the distribution of FLS2, enhanced the cellular response to flg22 and the sensing of latter activated CK response and there was links to the actin cytoskeleton and the late endosomes in cellular trafficking.

Some recommended modifications which can be discussed further are:

Page 2 line 89, please write the fungal species in full as it is the first time introduced in this section.

For experiments shown in figures 1,2,3 and 5, 100 µM CK (6-BAP) was used. Is there a reason for that choice? What was observed at lower concentrations? As in figure 4, there are different concentrations tested to study the effect of ARA6 endosomes. Similarly in Figure 5, actin was affected but not tubulin by CK at that stated concentration? What about the concentrations used in figure 4 i.e. 0.1, 1 and 10 µM CK? Did the author observe any change with the application of flg22 on ARA6-GFP/FM4-64 localization or actin?

In figure 4, A and B are measurements specific to epidermal cells? Would there be a difference in different cell types within the roots?

In the method section, page 9 line 290 please include “nm” after “588-641”; and for line 292 please specify the dwell time i.e. seconds, ms etc. Would this be the same for all experiments which had transgenic labelled plants?

Author Response

Many thanks to the reviewer for the time and effort invested in our manuscript.

Page 2 line 89, please write the fungal species in full as it is the first time introduced in this section.

Corrected, thanks.

For experiments shown in figures 1,2,3 and 5, 100 µM CK (6-BAP) was used. Is there a reason for that choice? What was observed at lower concentrations? As in figure 4, there are different concentrations tested to study the effect of ARA6 endosomes. Similarly in Figure 5, actin was affected but not tubulin by CK at that stated concentration? What about the concentrations used in figure 4 i.e. 0.1, 1 and 10 µM CK? Did the author observe any change with the application of flg22 on ARA6-GFP/FM4-64 localization or actin?

Many thanks for these comments.

We normally use 100 µM CK in tomato and tobacco, and in defense related assays, to generate strong effects. For the cellular trafficking aspect, which is a general assay and was done in Arabidopsis, we wanted to examine the effect of different CK concentrations. With the exception of ARA6 compartment density, we did not observe a dose-response effect but rather, a significant effect and 1 µM and above, and no effect below 1 µM. We added discussion of this to the manuscript. Thus, we did not see a need to examine additional concentrations, and selected the concnetration used in other experiments, that gives a strong effects at the cellular level.

Flg22 is known to induce the entry of FLS2 to endosomes after 60-75 minutes of treatment, which is longer than what we did in our assays (30 minutes). Flg22-Induced FLS2-GFP Endosomes were shown to Co-localize to ARA6 Compartments, though there was no difference in the amount of ARA6 endosomes at time points between 30 minutes and 105 minutes after flg22 application (see: Beck M, Zhou J, Faulkner C, MacLean D, Robatzek S. Spatio-temporal cellular dynamics of the Arabidopsis flagellin receptor reveal activation status-dependent endosomal sorting. Plant Cell. 2012;24(10):4205-4219. doi:10.1105/tpc.112.100263). Our paper studies the effect of CK, and not flg22, on cellular compartments. Therefore, we did not assay the effect of flg22 on cellular compartments without CK treatment.

In figure 4, A and B are measurements specific to epidermal cells? Would there be a difference in different cell types within the roots?

Many thanks. This is the common methodology for studying trafficking in Arabidopsis root cells. Entrance of Fm-4-64 to cortex cells likely occurs in different time frames, if at all, and is not the subject of the current investigation.

In the method section, page 9 line 290 please include “nm” after “588-641”; and for line 292 please specify the dwell time i.e. seconds, ms etc. Would this be the same for all experiments which had transgenic labelled plants?

The information was added to the materials section. The pixel dwell time was the same for all images.

Reviewer 4 Report

In this study, the authors investigated effect of cytokinin (CK) on cellular trafficking and on the actin cytoskeleton in plant cells. They found that CK regulates actin distribution, endocytosis, and pattern recognition receptors (PRR) trafficking. They claimed that CK has an important dual role in plant growth and disease resistance. Some interesting data are shown in the paper. Here I would like to suggest some point to be improved.

Major points

1) Introduction L.104. Please explain about details of FLS2 and flg22. Also, spell out name of FLS2 and LeEIX2. The reviewer recommends adding a previous work in the text (Chinchilla et al., Nature 2007).

2) Please explain about FLS2 in other plant species as well as tomato.

3) Figure 3. The reviewer would like to know results of control without flg22.

4) Discussion L.211-214. These sentences can move to introduction section.

5) Can the authors provide a hypothetical model as one figure?

6) Please discuss about similarity and differences on effect of CK in plants and fungi deeply.

Minor points

1) L.48, 52, and 55. Cytokinin > CK. Please confirm them.

2) Fig. 1A and 2A. Is the scale bar 100 µM? It is 100 µm. In Fig. 4 and 5, the scale bars are described in the figure legend.

3) L.140 and 173. Transgenic A. thaliana root roots stably...? Is this correct?

4) L.180. Anova > ANOVA

5) L.255 and 259. Delete a space between number and ˚C.

6) L.132. Us? We? Subject of the sentence.

7) References. There are a lot of mistakes in the text.

No. 3: The Plant Cell > Plant Cell

No. 18: Molecular plant pathology > Molecular Plant Pathology

No. 28: Insert 'USA' after Proceedings of the National Academy of Sciences.

No. 30: Plant physiology > Plant Physiology

No. 34: Proc Natl Acad Sci USA? Different from No. 28.

Gene name and scientific name should be written in italic.

Please check all the references again.

Author Response

Many thanks to the reviewer for the time and effort invested in our manuscript.

Major points

1) Introduction L.104. Please explain about details of FLS2 and flg22. Also, spell out name of FLS2 and LeEIX2. The reviewer recommends adding a previous work in the text (Chinchilla et al., Nature 2007).

The information was added, thanks.

2) Please explain about FLS2 in other plant species as well as tomato.

The information was added, thanks.

3) Figure 3. The reviewer would like to know results of control without flg22.

Controls were added to the assay and are now included in Figure 3. CK does have a small but significant effect on ROS production on its own, as was previously reported. This information was added to the text.

4) Discussion L.211-214. These sentences can move to introduction section.

Respectfully disagree. The idea on commonalities between developmental and defense mechanisms mediated by CK is, we believe, a point for the discussion.

5) Can the authors provide a hypothetical model as one figure?

A model is now included in Figure 6 and discussed in the discussion. Many thanks for this comment.

6) Please discuss about similarity and differences on effect of CK in plants and fungi deeply.

Text was added to the discussion, as suggested.

Minor points

1) L.48, 52, and 55. Cytokinin > CK. Please confirm them.

Corrected, thanks.

2) Fig. 1A and 2A. Is the scale bar 100 µM? It is 100 µm. In Fig. 4 and 5, the scale bars are described in the figure legend.

Corrected, thanks.

3) L.140 and 173. Transgenic A. thaliana root roots stably...? Is this correct?

Corrected, thanks.

4) L.180. Anova > ANOVA

Corrected, thanks.

5) L.255 and 259. Delete a space between number and ˚C.

Corrected, thanks.

6) L.132. Us? We? Subject of the sentence.

Corrected, thanks.

7) References. There are a lot of mistakes in the text.

No. 3: The Plant Cell > Plant Cell

No. 18: Molecular plant pathology > Molecular Plant Pathology

No. 28: Insert 'U

SA' after Proceedings of the National Academy of Sciences.

No. 30: Plant physiology > Plant Physiology

No. 34: Proc Natl Acad Sci USA? Different from No. 28.

Gene name and scientific name should be written in italic.

Please check all the references again.

We corrected the references, thanks. PNAS does not require USA in its title, so this was removed.

Round 2

Reviewer 1 Report

The manuscript is now worth to be published. The minor typos are shown in the attached annotated pdf file, the suggested changes are highlighted

Reviewer 4 Report

The paper has been improved according to the our comments.